# Ecological dependencies make remote reef fish communities most vulnerable to coral loss

Giovanni Strona [1]✉, Pieter S. A. Beck[2], Mar Cabeza [1], Simone Fattorini [3], François Guilhaumon[4,5], Fiorenza Micheli [6], Simone Montano[7,8], Otso Ovaskainen [1,9,10], Serge Planes[11,12], Joseph A. Veech[13] & Valeriano Parravicini [11]

Ecosystems face both local hazards, such as over-exploitation, and global hazards, such as climate change. Since the impact of local hazards attenuates with distance from humans, local extinction risk should decrease with remoteness, making faraway areas safe havens for biodiversity. However, isolation and reduced anthropogenic disturbance may increase ecological specialization in remote communities, and hence their vulnerability to secondary effects of diversity loss propagating through networks of interacting species. We show this to be true for reef fish communities across the globe. An increase in fish-coral dependency with the distance of coral reefs from human settlements, paired with the far-reaching impacts of global hazards, increases the risk of fish species loss, counteracting the benefits of remoteness. Hotspots of fish risk from fish-coral dependency are distinct from those caused by direct human impacts, increasing the number of risk hotspots by ~30% globally. These findings might apply to other ecosystems on Earth and depict a world where no place, no matter how remote, is safe for biodiversity, calling for a reconsideration of global conservation priorities.

[1] Organismal and Evolutionary Biology Research Programme, Faculty of Biological and Environmental Sciences, University of Helsinki, Helsinki, Finland. [2] European Commission, Joint Research Centre (JRC), Ispra, Italy. [3] Department of Life, Health & Environmental Sciences, University of L'Aquila, L'Aquila, Italy. [4] MARBEC, IRD, CNRS, Univ. Montpellier, Ifremer, France. [5] IRD, Saint-Denis de la Réunion, France. [6] Hopkins Marine Station and Stanford Center for Ocean Solutions, Stanford University, Pacific Grove, CA 93950, USA. [7] Department of Earth and Environmental Sciences (DISAT), University of Milan—Bicocca, Milan, Italy. [8] MaRHE Center (Marine Research and High Education Center), Magoodhoo Island, Faafu Atoll, Republic of Maldives. [9] Department of Biological and Environmental Science, University of Jyväskylä, P.O. Box 35 (Survontie 9C), FI-40014 Jyväskylä, Finland. [10] Department of Biology, Centre for Biodiversity Dynamics, Norwegian University of Science and Technology, Trondheim N-7491, Norway. [11] PSL Research University: EPHE-UPVD-CNRS, USR 3278 CRIOBE, Université de Perpignan, 66860 Perpignan Cedex, France. [12] Laboratoire d'Excellence "CORAIL", EPHE, PSL Research University, UPVD, CNRS, USR 3278 CRIOBE, Moorea, French Polynesia. [13] Department of Biology, Texas State University, San Marcos, Texas 78666, USA. ✉email: giovanni.strona@helsinki.fi

The effects of human activities on our planet are so pervasive[1] that many denote the current epoch as the Anthropocene[2]. In these challenging times for biodiversity, species face extinction[3,4], and ecosystems deteriorate under the synergic influence of global hazards (such as climate change) and local human stressors (such as overexploitation)[5,6]. Since global hazards act indeed globally, while local ones are associated with proximity to human activities, their combined effect is expected to decrease with the remoteness of the local ecosystem (Fig. 1a). Therefore, pristine and isolated ecosystems—sometimes referred to as "wilderness areas"—are considered sanctuaries that have the potential to preserve nature during the current and future biodiversity crises[7].

However, local anthropogenic disturbances can favour generalist species over specialized ones[8–10], as corroborated by previous work showing a positive relationship between the degree of ecological specialization and time with no disturbances in in-silico ecological networks[11]. In addition, due to the reduced in-flow of individuals into communities, we might also expect a higher specialization of ecological interactions in isolated habitats[12]. Specialized consumers can be more efficient in using their (few) resources when these are available but have, in principle, a higher co-extinction risk than generalist species[13,14]. Thus, while specialization increases ecological networks' robustness to species loss under stable environmental conditions, it also makes them more fragile towards potential cascading effects of primary extinctions (triggered, for example, by warming)[11]. Therefore, undisturbed and isolated communities should have many specialized interactions increasing their vulnerability to global change (Fig. 1a). Such an ecological mechanism depicts a component of risk which is distinct and adds up to that stemming from the increased chances of local extinction that species are experiencing in isolated habitats[15].

Here we test whether a positive relationship between ecological specialization/vulnerability and remoteness exists in natural systems, and whether the resulting increased risk of species loss in remote areas can question the common reliance on remote areas as biodiversity strongholds. For these goals, we focused on one of the most biologically diverse and socio-economically significant ecosystems on the planet, coral reefs, which, despite international attention and global protection programmes, continue to deteriorate under the influence of local human impacts (such as physical destruction and pollution) and the increasing effects of climate change (such as coral bleaching)[16–19]. By assessing the local dependency of fish assemblages on corals across the world's oceans, we show that the increase in the frequency and strength of fish-coral associations with distance from human settlements, combined with the global reach of coral bleaching, obliterate the benefits of remoteness on reef fish local extinction risk.

## Results and discussion
**Exploring the risk-remoteness relationship in reef fish**. We quantified remoteness as travel time to major cities[20,21] (Fig. 2a). This measure captures both the local impact of direct anthropogenic disturbances (Fig. 1b) and geographical isolation (Supplementary Fig. 1), being therefore well suited to test our hypotheses. Using a global dataset providing standardized measures of anthropogenic impacts on oceans[19], we quantified the cumulative risk of species loss for reef fish assemblages from local and global hazards. Local hazards stem from direct human activities (six impacts related to fishing activities plus light pollution, shipping, nutrient pollution, organic chemical pollution, and direct human impacts on coastal and near-coastal habitats). They decline with increasing remoteness from human settlements (Figs. 1b, 2b). Global hazards are related to global processes such

as ocean warming, ocean acidification and sea-level rise. They also decline with increasing remoteness but in a much weaker way (Figs. 1c, 2c). These patterns indicate that the necessary conditions for the risk-remoteness relationship to occur are met (Fig. 1a).

Given that we were able to demonstrate the necessary conditions empirically, we then addressed our primary questions. Specifically, we explored (i) the relationship between reef remoteness and strength of fish-coral ecological interactions; and (ii) the potential effect of such a relationship on the shape of the risk-remoteness relationship for reef fish. Such explorations required first assessing the degree of fish-coral dependency globally. The fish species known from literature to rely exclusively on corals for food or shelter represent only a fraction (~20%) of local coral reef fish diversity[22–25]. However, experimental evidence suggests that the loss of corals may affect more than a half of fish diversity[26], as also supported by recent statistical estimates[27]. This mismatch highlights that assessing fish assemblages' vulnerability to coral loss requires considering the dense network of elusive, direct, and indirect links[28] that create interaction pathways from coral to fish species.

To assess the influence of both direct and indirect coral-fish links on fish species persistence, we collected information on the global distribution and ecological traits of 9,143 fish species associated with coral reefs. We used these data and analytical approaches of previous studies[29–32] to identify potential trophic and habitat-related associations between corals and fish, and between prey and predatory fish species. We constructed local-scale networks of potential coral → fish → fish interactions (on a spatial grid of 1° × 1° covering 1761 reef localities worldwide) by combining previously published information on fish dependency on corals, spatial co-occurrences of species (accounting for species niche and biogeographical history), and the ecological traits of fish species. Finally, we quantified the dependency of fish assemblages on corals as the proportion of fish species in each locality (i.e., 1° × 1° cell in our grid) with direct or indirect links to corals within the local ecological network (Fig. 2f). This crucial step enables identifying indirect dependencies that would not have been apparent by just tallying coral dependent fish species from the literature. We found that the dependency of fish assemblages on corals increases with coral reefs' remoteness. These results support the remoteness-specialization hypothesis (Fig. 1f) and provide an important confirmation that the co-evolutionary mechanisms affecting the emergence of specialization in ecological networks identified by theoretical work[11,12] also apply to real-world systems. Furthermore, the average percentage of fish species identified as dependent on corals by our network approach (38% ± 10% s.d.) matches a recent global scale estimate obtained with a completely independent statistical model (41% ± 18% s.d.)[27], corroborating the idea that a world without corals might have half as many fish species.

We then decomposed the fish-coral dependency by distinguishing between fish directly associated with corals (i.e., having a minimum distance to corals in the network of one link) compared to fish indirectly linked to corals (i.e., having a minimum distance to corals of more than one link). We found that the relative importance of directly associated fish increases with remoteness (Fig. 3), which further strengthens the support for the hypothesis. Not only does the overall fish coral dependency increase with remoteness from a quantitative perspective, but the relative contribution of direct dependencies becomes stronger. Since we expect the effects of coral loss to be stronger on directly coral-associated fish than on indirectly associated fish, this result reinforces the idea that remote communities will be substantially more affected than accessible ones as the impacts of global change propagate across ecological networks. An extensive set of

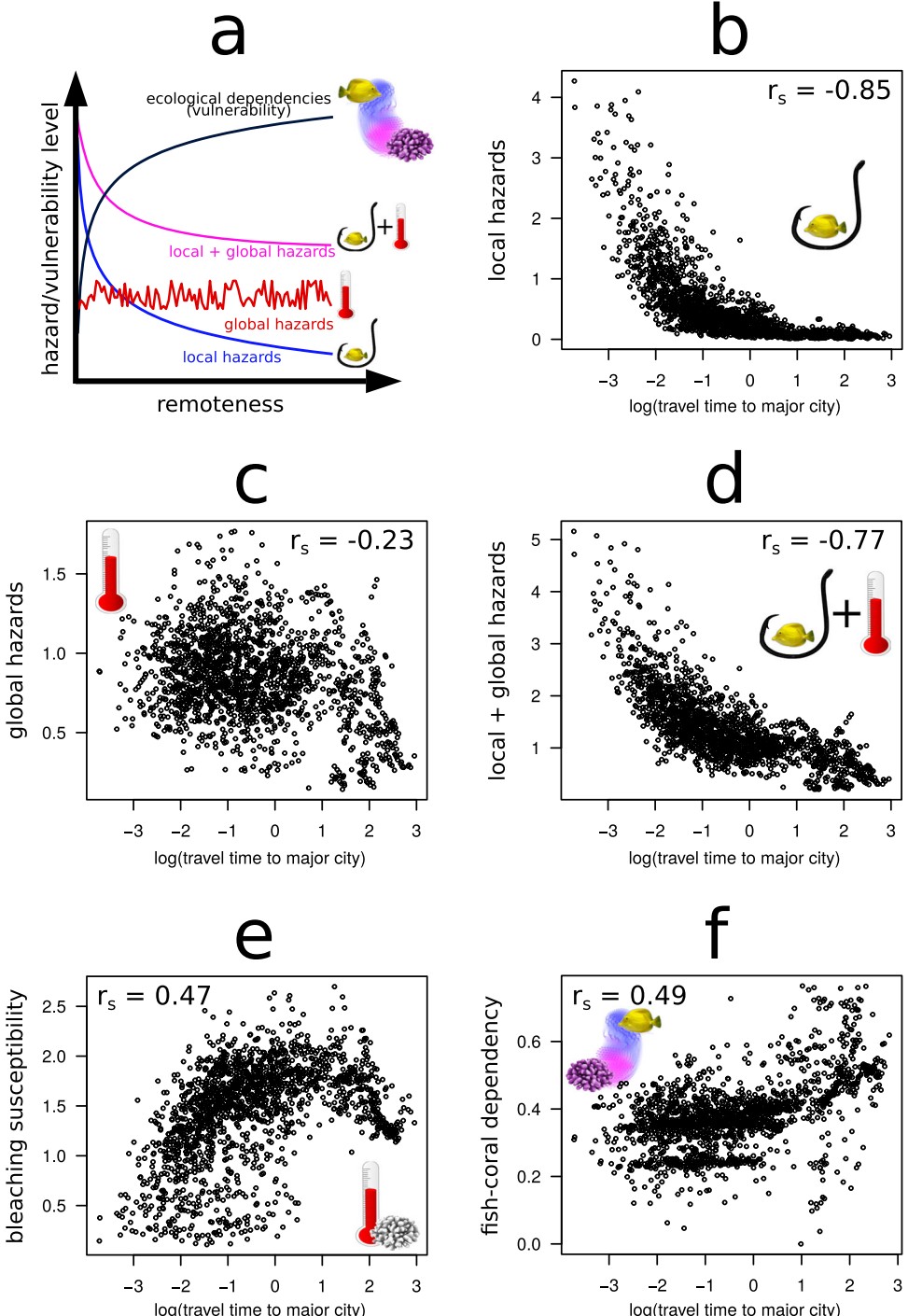

**Fig. 1 Theoretical and empirical relationships between remoteness vs local/global hazards and ecosystem vulnerability from ecological dependencies.**
**a** Theoretical expectation of a decrease in local and local + global hazards with remoteness, and a counteracting increase in ecosystem vulnerability due to ecological dependencies. **b** Comparison between reef remoteness, measured as travel time (in $\log_e$ transformed hours) from a reef locality to the closest major city[21], and local hazards (cumulative local impacts on reef localities for 2013, consisting of six impacts related to fishing activities, light pollution, shipping, nutrient pollution, organic chemical pollution, and direct human interactions on coastal and near-coastal habitats[19]). **c** Comparison between reef remoteness and global hazards (cumulative global impacts on reef localities for 2013, consisting of warming, acidification, and sea level rise[19]). **d** Comparison between reef remoteness and cumulative local + global impacts. **e** Comparison between reef remoteness and bleaching susceptibility quantified, for each reef locality, as the average bleaching alert level from 1985 to 2019. **f** Comparison between reef remoteness and fish-coral dependency (quantified as the fraction of fish diversity directly or indirectly connected to corals through a coral → fish → fish network path at 1761 reef localities at a resolution of 1° × 1°). For each relationship, we report the Spearman's rank correlation coefficient ($r_s$).

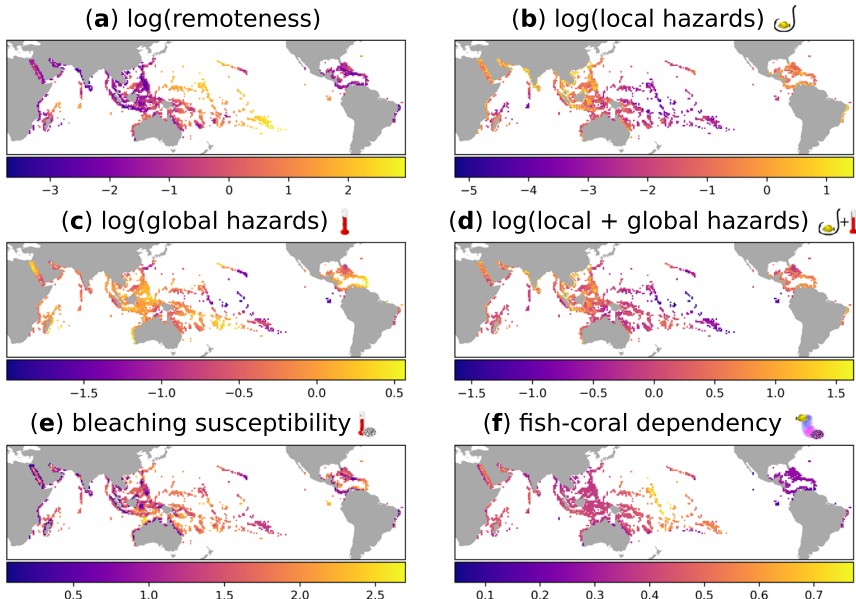

**Fig. 2 Global maps of reef remoteness, local and global hazards, bleaching susceptibility and fish-coral dependency. a** Global remoteness of coral reefs, quantified as travel time (in $\log_e$ transformed hours) from the target reef locality to the closest major city[21]. **b** $\log_e$ transformed local hazards (cumulative local impacts on reef localities for 2013, consisting of: six impacts related to fishing activities, light pollution, shipping, nutrient pollution, organic chemical pollution and direct human interactions on coastal and near-coastal habitats[19]). **c** $\log_e$ transformed global hazards (cumulative global impacts on reef localities for 2013, consisting of: warming, acidification and sea level rise[19]). **d** $\log_e$ transformed local + global hazards; **e** global bleaching susceptibility, quantified as the average bleaching alert level from 1985 to 2019. **f** Fish-coral dependency, quantified as the proportion of fish species that are directly or indirectly connected to corals through an identified coral → fish → fish network path at 1761 reef localities at a resolution of 1° × 1°.

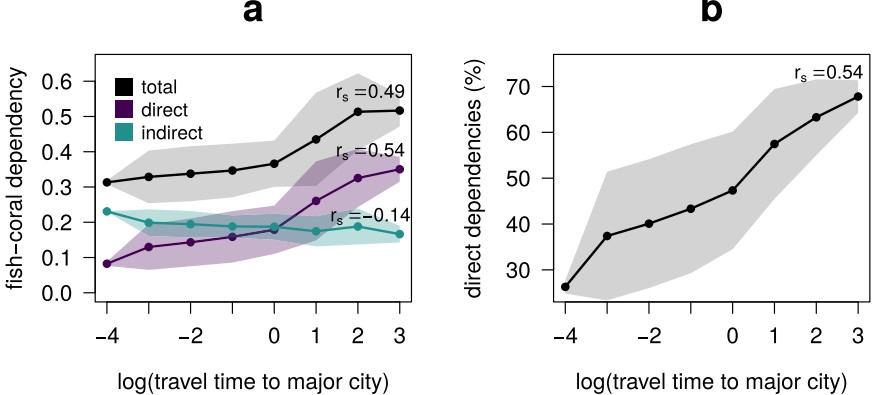

**Fig. 3 The relative contribution of direct fish-coral dependency increases with reef remoteness.** We decomposed the total fish-coral dependency (i.e. the total fraction of fish species having at least one path to corals in the local coral → fish → fish networks) by distinguishing between fish species having a minimum distance of 1 step (i.e. network link) to corals, and fish species having a minimum distance to corals >1 step. While the fraction of fish with direct associations with corals increases with remoteness, that of indirectly associated fish decreases (**a**). Thus, as we move away from human influence, the relative contribution of direct fish-coral dependency increases from 26 to 68% on average (**b**). The plots summarize the results obtained in 1761 reef localities at a resolution of 1° × 1°. Solid lines represent average values, while shaded areas represent standard deviations. The Spearman's rank correlation coefficients ($r_s$) were computed on the full set of results ($n = 1761$), and not on the averaged values. Remoteness of coral reefs was quantified as travel time (in $\log_e$ transformed hours) from the target reef locality to the closest major city[21].

sensitivity analyses confirm that these results are not affected by potential biases in the availability of information on fish ecology and distribution, nor are they driven by geographical variation in functional redundancy or species abundances (see "Methods" and Supplementary Fig. 2).

**Mapping fish risk hotspots**. The effect of global and local hazards and that of ecological dependencies show a striking spatial complementarity in determining global reef-fish risk. We mapped areas of high local + global hazards (falling in or above

the 70th percentile) as well as areas of high combined fish-coral dependency and bleaching susceptibility. The latter are reef localities in or above the 70th percentiles of both fish-coral dependency and bleaching susceptibility, and comprise 9.4% of reef localities (165 1° × 1° cells of our global reef map). Comparing the two maps reveals how only nine reef localities, or 0.5% of areas highly threatened by local and global hazards also have a high fish-coral dependency and bleaching susceptibility. Thus, when we consider as hotspots of risk all localities from either of the two maps, the total number of reef fish assemblages at risk

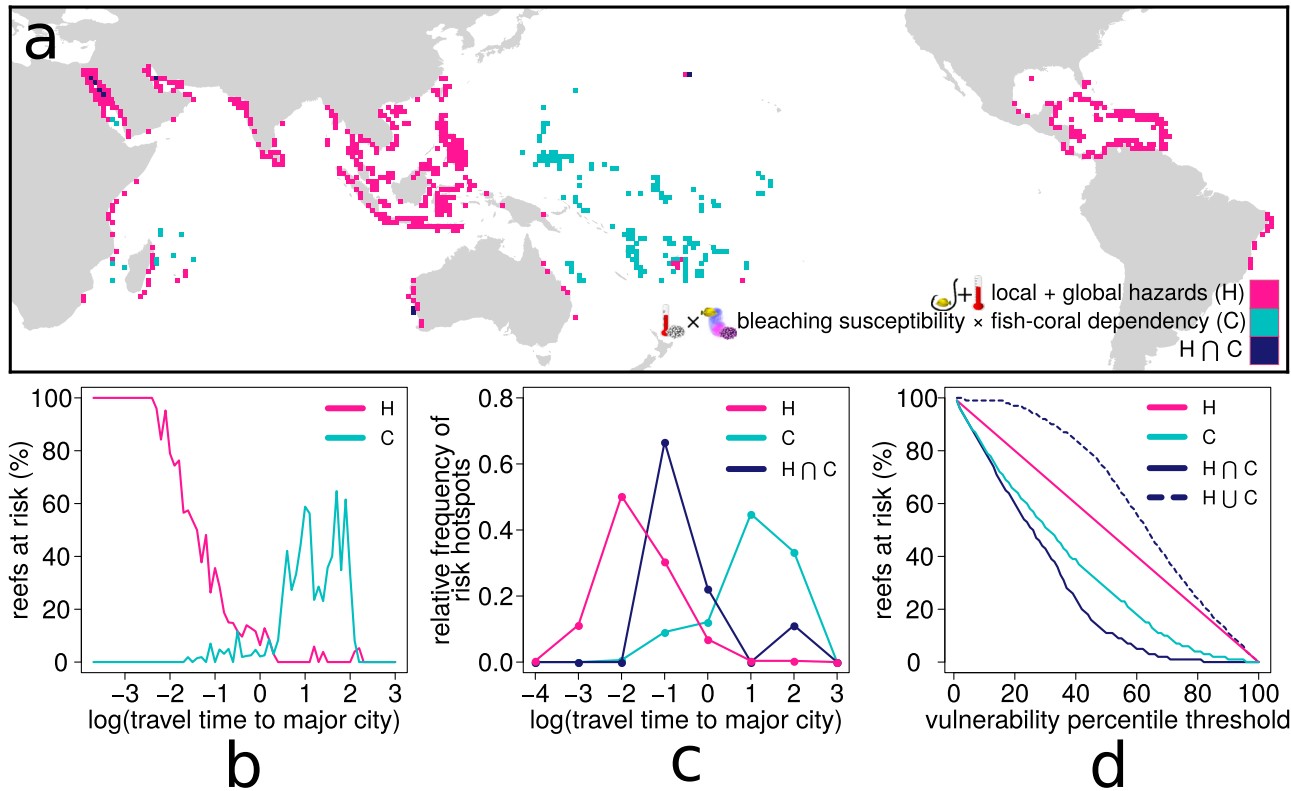

**Fig. 4 Spatial comparison between hot-spots of risk from local and global hazards vs. hotspots of risk from fish-coral dependency combined with bleaching risk. a** Magenta pixels are reef localities (at a resolution of 1° × 1°) falling above the 70th percentile of local+global hazards (based on 2013 cumulative human impacts on reef localities[19] as in Fig. 2d); cyan pixels are reef localities falling simultaneously above the 70th percentile of fish-coral dependency (fraction of fish diversity per reef locality directly or indirectly connected to corals through the coral → fish → fish network, as in Fig. 2f) and above the 70th percentile of bleaching susceptibility (quantified, for each reef locality, as the average bleaching alert level from 1985 to 2019 as in Fig. 2e); dark blue pixels are reef localities falling in both of the previous categories. **b** Percentage of reef localities worldwide where the fish community is put at risk by either local+global hazards (magenta line) or by fish-coral dependency combined with bleaching susceptibility (cyan line) for increasing values of remoteness, quantified as travel time (in $\log_e$ transformed hours) from the target reef locality to the closest major city[21]. **c** Frequency of reef risk hotspots from either local+global hazards (magenta line), fish-coral dependency combined with bleaching susceptibility (cyan line), or both (dark blue line), for increasing values of remoteness (frequency relative to the respective total number of risk hotspots; data were pooled to the first decimal digit of remoteness). **d** Percentage of reef localities worldwide where the fish community is put at risk by either local + global hazards (magenta line), fish-coral dependency combined with bleaching susceptibility (cyan line), at least one of these two sources of risk (dashed dark blue line), or both (continuous dark blue line), for different percentile thresholds used to identify hotspots. The thresholds were identified (and applied) independently for local + global hazards, fish-coral dependency and bleaching susceptibility.

increases by 29%, from 535 to 691 reef localities (39.2% of reefs) (Fig. 4). Further, our study reveals that the fish communities on some of the most remote coral reefs are at relatively high risk of local species extinction (Figs. 2 and 4).

Thus, the effects of local and global hazards in reef fish assemblages and those of ecological dependencies combined with bleaching vulnerability show a remarkable complementarity. Many areas that are not hotspots of risk from global or local hazards are potential hotspots of risk due to ecological network fragility and vice versa. This pattern is a strong warning that the ongoing biodiversity crisis is truly global and that distance from human influence does not guarantee safety. In turn, it highlights a profound need to account for ecological dependencies when assessing the risk global change poses to particular species.

**Accounting for ecological dependencies in risk assessment**. The very different nature of the risk sources makes exploring the potential effect of the remoteness-specialization relationship on global risk projections challenging. Here, the risk assessment framework provided by IPCC's fifth assessment report—which

quantifies risk as the combination of vulnerability, exposure, and hazard[5]—might provide a formal layout to tackle the challenge.

As a proof of concept, we devised an equation which quantifies risk by combining additively global and local hazards with the effect of ecological dependencies as applied to our fish-coral case study. To include the effect of ecological dependencies, we had to identify a potential "trigger" capable of transforming the vulnerability stemming from fish-coral dependency into an additional component of local risk. An obvious trigger is local susceptibility to bleaching events[16–18], which we identified based on bleaching alert level data from 1985 to 2019 (Fig. 2e; see "Methods" for details). Bleaching is a global hazard (in that its cause does not originate from a single point source) that can have local effects. Bleaching susceptibility can indicate the probability of local coral mortality and loss. Combining bleaching susceptibility with the local estimate of fish-coral dependency (from the network analysis) quantifies, therefore, a local risk for fish communities stemming from the bottom-up effects of coral loss across coral-fish networks.

Depending on the different weights assigned to either the risk component stemming from global and local hazards or to the one

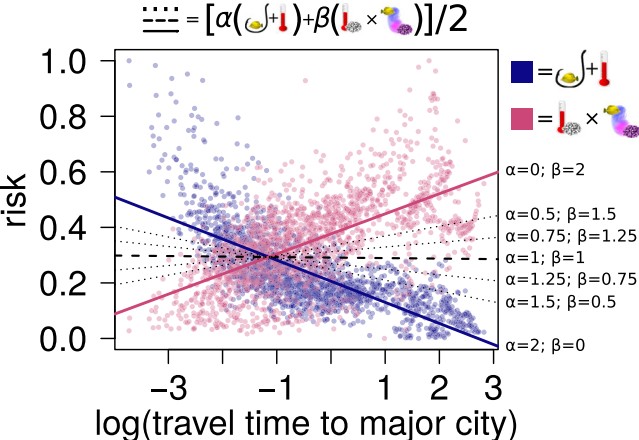

**Fig. 5 Fish-coral dependency modifies the risk-remoteness relationship.** Coral reef remoteness was quantified as travel time (in $\log_e$ transformed hours) from the target reef locality to the closest major city[21]. The blue dots represent risk quantified as the sum of threats from local + global hazards on reefs (as in Fig. 2d), while magenta dots represent risk quantified as bleaching susceptibility × fish-coral dependency. Both components of risk (i.e., local + global hazards and bleaching susceptibility × fish-coral dependency) were rescaled between 0 and 1. The two rescaled risks components are then combined into a single risk assessment equation where risk = [$\alpha$ (local + global hazards) + $\beta$ (bleaching susceptibility × fish-coral dependency)]/2. The lines in the plot represent the slopes of the trend lines from different parametrizations of the risk equation. When equal weight is given to the two risk components, risk remains almost constant across remoteness values (trend line slope = −0.002, black dashed line).

stemming from ecological dependencies (i.e., the $\alpha$ and $\beta$ terms in Eq. 4) we can identify different patterns for the risk-remoteness relationship. The two extremes correspond to the risk emerging from, alternatively, only local and global hazards, or only ecological dependencies triggered by local bleaching susceptibility (Fig. 5). However, under the parsimonious assumption that both sources of risk are equally important for fish species (i.e., for example, that a coral dependent fish species would be equally threatened by mass coral mortality as by overfishing) the risk-remoteness relationship becomes flat, providing a strong argument that distance from humans does not make a fish community any safer.

**The risk-remoteness relationship in global conservation.** With reef fish providing protein to half a billion people worldwide[33] and the critical importance of fish for addressing micronutrient deficiencies[34], our results have profound societal implications; remote coral reefs won't be able to compensate for the losses of coral and fish species directly impacted by human activities, threatening the livelihoods of millions. Also, our study reveals an essential macroecological and eco-evolutionary mechanism that might dramatically amplify risks from global change in natural systems.

The risk patterns observed for reef fish communities suggest that our already disconcerting projections about biosphere fragility might be overly optimistic. Moreover, the results of our study temper any hopes that, by protecting wilderness areas, we safeguard biodiversity vaults that can withstand the past and ongoing environmental destruction and changes brought by the Anthropocene. Therefore, aggressively addressing global hazards while supporting local management and conservation at both intensely used and remote locations emerges as the only hope to reverse the current biodiversity crisis.

## Methods

**Fish distribution.** We rasterized a detailed reef distribution vector map[35] at 5 × 5 latitude/longitude degrees (by considering as reef area each cell in the raster intersecting a polygon in the original shapefile). We collected all the occurrences of fish species intersecting the rasterized reef area from both the Ocean Biogeographic Information System[36] and the Global Biodiversity Information Facility[37]. We used taxonomic and biogeographical (i.e., latitudinal/longitudinal extremes for a given species) information from FishBase[38] to exclude potential incorrect occurrences (i.e., all the records falling outside the known species ranges). We also restricted the list to all the species for which FishBase provided relevant ecological information (as these were needed to evaluate prey-predator species interactions and identify indirect links between fish species and coral, see below). The filtered list comprises 9143 fish species.

For these species, we used occurrence data to generate species ranges. For this, we used the $\alpha$-hull procedure[39], but instead of pre-selecting an $\alpha$ parameter and using it for all species, we developed a procedure to obtain conservative species ranges while including most of the known occurrences. First, we selected a very small $\alpha$ (0.001), to obtain a hull including most of the occurrences. Then, we progressively incremented $\alpha$ in small amounts (0.005) by computing, for each increment, the ratio between the relative reduction in the resulting hull area (in respect to the previous hull), and the relative reduction of occurrences included in the hull (in respect to the total number of available occurrences for the target species). We stopped increasing $\alpha$ when the ratio became <10. This procedure ensured that only isolated sites far from the core distribution of a species were excluded, while the range was stretched as much as possible around known occurrences.

After delineating ranges for each species, we rasterized the reef vector map at a higher resolution (1 × 1 latitude/longitude degree) and used it as a reference layer to extract fish occurrences at each reef location. This resolution is finer than that used by other global studies on reef fish diversity and distribution[40,41]. We took the 1° × 1° reef raster as the reference grid in all subsequent analyses and spatial interpolations, considering all the reef cells hosting at least five fish species (n = 1761).

**Fish distribution validation.** To validate the fish distribution data, we compared them with a smaller independent dataset (GASPAR) providing fish occurrences for 196 globally distributed reef localities[42], which we rasterized against the same reference grid used for our fish and coral distribution data. Because this dataset is based on comprehensive check-lists, its information can be considered as ascertained presence-absence data. Thus, we compared our list of fish occurrences (at one degree) in each cell where data from the GASPAR dataset were also available, computing true skill statistics score as $TSS = [(a \times d) - (b \times c)]/[(a + c) \times (b + d)]$, with $a$ being predicted & observed occurrences; $b$ being predicted, but not observed occurrences; $c$ being observed but not predicted occurrences; and $d$ being not observed and not predicted occurrences. We obtained a median TSS of 0.53, with a median sensitivity (the proportion of correctly predicted presences) of 0.60, and a median specificity (the proportion of correctly predicted absences) of 0.96, indicating that our mapped ranges were sufficiently conservative and rarely generated false presences. Finally, given that we were analysing coral reef fishes, we excluded a few grid cells where our methods returned no fish species.

**Environmental data.** We obtained environmental data (surface temperature, salinity, pH, and total chlorophyll as a proxy for productivity) at a spatial resolution of 5 arcmin from Bio-ORACLE v2.0[43], and we upscaled these data on the reference reef grid (averaging the variable values in each 1 × 1 latitude/longitude degree grid).

**Human impact.** As a measure of human impact on reef localities, we used the 14 cumulative human impact layers (for 2013)[19] available at https://doi.org/10.5063/F12B8WBS. For the purposes of our analysis, we categorized them into "local hazards" stemming from direct human impacts (specifically, six impact layers related to fishing activities plus light pollution, shipping, nutrient pollution, organic chemical pollution, and direct human interactions on coastal and near-coastal habitats, such as trampling); and "global hazards" related to planetary wide processes (warming, acidification and sea level rise). The original dataset has a resolution of 1 km$^2$ and was therefore upscaled on the reference reef grid (averaging the variable values in each 1 × 1 latitude/longitude degree grid).

**Time travel to cities.** We quantified the "remoteness" of each reef locality in terms of travel time (based on the fastest possible local means of terrestrial and aquatic transportation, hence excluding air travel) to the closest human settlement. For this, we used the procedure described in Weiss et al.[21] which consists of first combining information on land types and use, topography, distribution of roads and railways, position of national borders to derive a friction surface raster map indicating the average speed at which humans can travel through each pixel; and then applying an algorithm to identify the least costly paths (i.e. those requiring the shortest travel time) from each pixel to a target locality (e.g. a city)[21].

The original publication[21] provides a global map of accessibility that does not include water localities, which is clearly problematic for reefs. We therefore produced a new map of travel time (in hours) including also water pixels (at the same resolution of Weiss et al.[21], i.e. 1 km$^2$) by using their friction map, the same

layer of human urban centre (the 'high-density centres' variant of the Global Human Settlements[44]) and the same cost distance algorithm (cumulative cost distance, which we computed using SAGA gis[45]). Then, we upscaled the high-resolution map on our grid of $1 \times 1$ degree reef localities (computing the mean accessibility per each $1 \times 1$ degree cell).

**Bleaching susceptibility**. We downloaded annual layers reporting maximum bleaching alert level at the global scale and at a resolution of 50 km from 1985 to 2019[46]. Alert levels range from 0 (no stress) to 4 (mortality likely). We upscaled each layer on the reef reference grid (averaging alert level data) and computed an index of bleaching susceptibility as the average of recorded alert level in each coral reef pixel of the reference raster.

**Building ecological networks of fish → fish interactions**. We built networks of fish → fish interactions by using a multi-step procedure. (1) We generated a model capable of predicting the probability of occurrence of a prey-predator interaction between two given fish species based on some of their functional and ecological traits. For this, we obtained information on fish body size, trophic level, minimum and maximum depth, and habitat preference for 17,722 fish species from FishBase[38], OBIS[36] and GBIF[37] (from the latter two sources, we specifically derived complementary data on species depth occurrences, which we used to fill in gaps in FishBase). We combined this information with a large dataset of known prey-predator interactions assembled from the Global Biotic Interactions dataset, GLOBI[47]. After filtering GLOBI according to the set of species with available ecological information and removing replicated records, we obtained 11,188 individual prey-predator pairs (for a total of 2643 species). We then identified an identical number of absences (pairs of species not interacting, and hence not having a link in the network). GLOBI includes only observed interactions, while it does not provide explicit information on non-interacting species. Although one can ideally generate a list of absences by sampling from all pairwise combinations of species not listed by GLOBI, this procedure might lead to the mislabelling of an actual prey-predator pair as a non-interacting pair simply because the species combination is missing from the database. To reduce this risk and generate "reliable" pseudo absences (that is, truly representative of associations not possible in the real world), we used a stochastic approach where we sampled species pairs at random from all possible species combinations not present in GLOBI with the important addition of two constraints; namely, the prey needed to be at least 30% larger than the predator and/or the predator needed to have a trophic level ≤3.0 (according to FishBase trophic classification).

(2) We then used a random forest classifier (a machine learning technique; we used the Python package *Scikit-learn*[48]) where the dependent variable was the presence or (pseudo) absence of interactions, and the independent variables were prey and predator traits (prey body size, prey trophic level, prey min and max depth and eight dummy variables for habitat; and the same variables for predator, for a total of 24 independent variables). We first explored the ability of the model by training it on a random subsample (50%) of the dataset (including true presences and pseudo absences), and then testing it on the remaining fraction. The model performed very well, being capable of predicting observed (true positives) and unobserved interactions (true negatives) in the testing set with an exceptional precision and accuracy (TSS = 0.93; type I error rate = 0.05; type II error rate = 0.02). After this first exploration, we used the full dataset to train the model to be used on the actual data. Out-of-bag validation score in the final model based on the complete dataset was >0.97.

The random forest predictor was used to assess the probability of trophic interaction between a large list of potential interactions generated by combining all fish species from our reef fish occurrence dataset known to rely mainly or exclusively on fish for their survival (i.e. "true piscivores", FishBase trophic level > 3.5), with all the fish in the dataset. The full list included 31,768,450 potential interactions, that we reduced to 6,721,450 interactions by keeping only the interacting pairs identified by the random forest classifier with a probability ≥0.9.

(3) If the ecological dependency between two species is actually manifested then the two species must obviously co-occur at some locations, and vice-versa, co-occurrence is a necessary pre-requisite for an ecological dependency. Following this logic, we took a final, additional step to further filter and improve the fish → fish interaction list. In particular, we quantified the tendency for species to co-occur in the same locality as one potential proxy layer for species interactions, complementary to our other approaches. There are various factors that can affect the co-occurrence of two species. In a simplification, this can emerge from stochasticity, shared environmental requirements, shared evolutionary history, and ecological dependencies. We attempted to disentangle the effect of the last factor from the first three.

For each target species pair, we computed overlap in distribution as the raw number of reef localities where both target species were found. Then, we compared this number with the null expectation obtained by randomizing the distribution of species occurrences across reef localities. We designed a null model accounting for randomness, species niche and biogeographical history, and hence randomizing the occurrence of species only within areas where they could have possibly occurred according to environmental conditions and biogeographical factors (e.g., in the absence of hard or soft barriers). To implement the null model, we first excluded from the list of potential localities all the areas outside the biogeographical regions where the target species had been recorded, with regions identified according to

Spalding et al.[49]. Then, within the remaining areas, we identified all the reef localities with climate envelopes favourable to target species survival. For this, we identified the min and max of major environmental drivers (mean annual surface temperature, salinity, pH) where the target species occurred, and then we identified all the localities with conditions not exceeding these limits. We generated, for each pairwise species comparison, one thousand randomized sets of species occurrences by rearranging randomly species occurrence within all suitable localities. We quantified co-occurrence between the species pair in each random scenario. Finally, we compared the observed co-occurrence with the random co-occurrences, computing a *p*-value as the fraction of null models with co-occurrence identical or higher than the observed one. We kept only the pairs with a *p*-value < 0.05. This further reduced the fish → fish list to 1,365,863 interactions. We used the networks to build site-specific networks interactions in all 1° × 1° reef localities of our reference grid, by filtering it according to local fish species diversity.

**Measuring fish-coral dependency**. We compiled from literature[22–25] a list of fish species known to be associated with corals, in terms of habitat and/or trophic specialization. This list includes 44% of the fish species we used in our analysis (4040/9,143). As above, we used the known associations (or lack thereof) in the dataset to identify coral dependency in the unassessed fish. For this, we trained two independent random forest classifiers (again using the Python package *Scikit-learn*[48]), one to model generic habitat associations, and the other one to model corallivory. In both models, the dependent variable was the presence/absence of coral-association, and the independent variables were the same ecological features used to predict fish → fish trophic interactions (i.e. prey body size, prey trophic level, prey min and max depth and eight dummy variables for habitat), plus an additional variable quantifying the fraction of documented coral-associated species in the family of the target fish. Both models showed high precision and accuracy (with a TSS of 0.57 for the habitat association model, and of 0.81 for the corallivory model). Combining the list of coral dependent species from literature ($n = 897$) with our model predictions ($n = 356$) yielded a total of 1253 fish species.

We linked all the coral-dependent species in the local fish → fish networks to a symbolic "coral" node. Then, we quantified the overall dependency of fish assemblages on corals in each reef locality as the fraction of fish having at least one (unidirectional) path to corals across network links. We opted for this simple and intuitive measure after finding it produced virtually identical results to several, more complex, measures of fish-coral dependency that we explored (such as weighted and unweighted network distance between individual fish species and coral genera, and dependency values estimated using co-extinction simulations[50]). For each network, we also quantified, separately, the fraction of fish species directly associated with corals (i.e., having a minimum distance to corals in the network of one link) and indirectly associated with corals (i.e. having a minimum distance to corals of more than one link).

**Risk assessment framework**. Following the definitions from the IPCC's fifth assessment report, we separate vulnerability (combination of sensitivity and adaptive capacity) from exposure to an extrinsic forcing agent ('hazard'). Then we quantify risk as the combination of vulnerability, exposure, and hazard[5].

Assuming, for illustrative purposes, a combined linear effect of local and global hazards on the risk experienced by a target system, we can model the latter ($R$) as:

$$R = E \times (H_{local} \times V_{local} + H_{global} \times V_{global}), \quad (1)$$

with $E$ being exposure, and $H_{local}$, $H_{global}$, $V_{local}$ and $V_{global}$ being local and global hazards and their respective vulnerabilities. If we then focus on average per-species risk, and assume no relationship between a system's remoteness and its intrinsic vulnerability to local and global hazards, we can further simplify the equation by setting $E$, $V_{local}$ and $V_{global}$ to 1:

$$R = H_{local} + H_{global} \quad (2)$$

To account for the effect of the expected increase in ecological dependencies with remoteness[8] in the illustrative risk assessment model described by Eq. (2), we can add one term to quantify the combined effect of the vulnerabilities emerging from ecological dependencies combined with the exposure to relevant hazards capable of exploiting such vulnerabilities and triggering cascading effects through interaction links ("triggers"):

$$R = [\alpha(H_{local} + H_{global}) + \beta(\text{ecological dependency} \times \text{triggers})]/2 \quad (3)$$

Here, $\alpha$ and $\beta$ are weights that can be used to modulate the relative importance of the two risk components (impacts from humans and global change vs ecological dependencies). Assuming that both risk components are rescaled in [0,1], to keep $R$ in [0,1], we need to set $0 \le \alpha \le 2$ and $\beta = 2 - \alpha$.

**Applying the risk assessment framework to reef fish communities**. We modelled the local risk of a reef fish community (in each 1° × 1° grid cells in the reef reference raster) using two different approaches. First, we quantified the risk as originating from the sum of local and global hazards (Eq. (2)), where local and global hazards refer to the human impact layers[19], as described in the "Human impact" section above. Then, we re-assessed risk for each reef fish assemblage when accounting also for the risk component possibly deriving from ecological

(fish-coral) dependencies combined with a relevant hazard (e.g., death of coral species due to bleaching) capable of triggering cascading effects across species interaction links by adapting Eq. (3):

$$R = [\alpha(H_{local} + H_{global}) + \beta(\text{coral dependency} \times \text{coral bleaching susceptibility})]/2$$
(4)

Fish-coral dependency and coral bleaching susceptibility were assessed as described in the sections above. To make the different components of risk comparable, prior to computing risk, we rescaled both local + global hazards and fish-coral dependency × coral bleaching susceptibility between 0 and 1 across all reef localities. We did the same for the two sets of risk assessment values obtained using either Eqs. (1) or (2) (to permit direct comparison between the shapes of the risk-remoteness relationships).

Both equations ideally provide the average risk of a species in a given locality, that is they assume exposure = 1. Also, they assume that the average local degree of vulnerability towards either local or global hazard is constant among localities; therefore, the respective vulnerability terms can be removed from the risk equations given that they are constants which would affect each locality the same. See the "Potential caveats in the risk assessment equations" section below for additional discussion on these issues.

**Assumptions of the risk assessment equations.** In this study we demonstrated how the framework of environmental risk assessment could incorporate species dependencies to more thoroughly examine the relationship between risk and remoteness. The proposed risk assessment equations are not intended to provide a definitive global risk assessment of reef fish assemblages. Instead, they are functional to assessing if, and to what degree, the risk component stemming from ecological dependencies can affect the expected relationship between risk and remoteness. The exact form of the equations is not overly important. In our equations we assumed constant vulnerability of fish assemblages to local and global hazards. That is, we ignored hazard-specific vulnerabilities. Although fish on coral reefs are likely vulnerable to the various hazards to different extents, modelling this amount of complexity would be extremely difficult. Considering the multiplicity of hazards per locality, and their potential complex interactions, it would be extremely challenging to obtain precise and realistic values for each of them to test our assumptions. However, we were able to compile several proxies of potential vulnerability to some of the main hazards, and in particular we computed the average vulnerability to fishing for all fish species in each reef locality, using the vulnerability measure provided by FishBase and based on the method by Cheung et al.[51]. Based on geographic distributions of the species, we determined the temperature, pH, and organic matter limits for each species, and then we used these data as indicators of each species potential tolerance to changes in temperature, acidification and organic pollution. Based on species habitat preference as defined by FishBase, we determined the fraction of demersal, benthopelagic, and coral associated species, as likely more affected by direct human disturbances (such as trampling); and the proportion of pelagic fish as potentially affected by shipping. We then compared those vulnerability proxies with remoteness, finding no strong relationships which would need to be incorporated into the risk equations (Supplementary Fig. 3).

Then, we explored if our results held when exposure was taken into account (i.e., projecting the average per-species risk to the full fish assemblages). Exposure is a typical parameter involved in environmental risk assessment. For this, we multiplied the risk for the (log_e-transformed) corresponding fish diversity. The observed patterns (Supplementary Fig. 4) were consistent with those relative to average species risk, which means that our conclusions scale up to fish assemblages. Again, the results of our study do not provide absolute estimates of risk for any of the fish species or coral reefs. However, with further research, we believe such estimates could be realistically obtained given sufficient species-specific data and more information about how the detrimental effects of each hazard are manifested.

**Sensitivity analyses.** We performed various analyses to check the robustness of our results and conclusions against potential biases stemming from data availability. In particular, we focused on potential relationships between the quality and quantity of information on species ecology and distribution, and remoteness. First, we checked for unequal distribution of sampling effort, under the hypothesis that remote localities could be less investigated than those close to human settlements. A comparison between the number of fish records available from OBIS[36] and GBIF[37] vs remoteness across all 1° × 1° reef localities revealed that this is not the case, with sampling effort remaining relatively high across all localities regardless of remoteness (Supplementary Fig. 2a, $R^2 = 0.0008$).

We then explored whether the availability and quality of the ecological information we used in our analyses decreased with remoteness. For this we evaluated how the TSS values obtained from the comparison between the species ranges devised with our procedure and independent species distribution data from the GASPAR dataset[42] varied across reef localities with remoteness. We found no relationship (Supplementary Fig. 2b, $R^2 = 0.0292$). We also looked at the individual species TSS values obtained by comparing the distribution of a target species devised by our procedure with that according to the GASPAR dataset. Consistently with the previous result, we found no pattern linking the average of local species'

TSS values to remoteness (Supplementary Fig. 2c, $R^2 = 0.0001$). We also explored whether remoteness affected negatively the fraction of species (for which we had distributional data) to be discarded in each locality due to the lack of the ecological information needed in our analyses. Again, the analysis revealed no effect of remoteness on data availability (Supplementary Fig. 2d, $R^2 = 0.0992$).

Another potential question arising from our conclusions is whether they would still be valid when species abundances are considered alongside species diversity. To explore this issue, we tested whether the relative abundance of coral-dependent fish changes with remoteness using all the data available from the Reef Life Survey (RLS) dataset[52]. Finding that coral dependent fish become less abundant as remoteness increases would weaken our results, as the increasing species-level vulnerability stemming from coral dependency would be counterbalanced by the reduction in the overall number of individuals threatened by coral loss. This is not the case. On average, coral associated fish are more abundant than the other species (with an average number of individuals per survey of 782 for coral associated species vs 658 for non associated species). More importantly, the local proportion of associated individuals is unaffected by remoteness (Supplementary Fig. 2e, $R^2 = 0.0002$).

Finally, we tested whether our results could be driven or confounded by a potential relationship between functional redundancy and remoteness. We quantified functional redundancy in each locality as one minus the ratio between the number of unique functional entities and total species richness. We identified functional entities using the method and functional diversity datasets as in Mouillot et al.[53]. We found no relationship between functional redundancy and remoteness (Supplementary Fig. 2f, $R^2 = 0.0042$).

**Reporting summary.** Further information on research design is available in the Nature Research Reporting Summary linked to this article.

## Data availability

All the data used in the analysis are freely available online from the sources listed in the Method section, and particularly: (1) reef distribution map: http://data.unep-wcmc.org/datasets/1; (2) fish occurrence data: https://obis.org/; and GBIF (Actinopterygii, https://doi.org/10.15468/dl.k6vam4; Elasmobranchii, https://doi.org/10.15468/dl.pu4tcx; Holocephali, https://doi.org/10.15468/dl.npckhm; Sarcopterygii, https://doi.org/10.15468/dl.huzujv); (3) fish ecology data: http://www.fishbase.org; (4) ocean impact layers: https://doi.org/10.5063/F12B8WBS; (5) the friction surface map needed to compute accessibility: https://malariaatlas.org/research-project/accessibility-to-cities; (6) human settlement data: http://data.europa.eu/89h/jrc-ghsl-ghs_smod_pop_globe_r2016a; (7) bleaching alert data: https://coralreefwatch.noaa.gov; (8) environmental layers: https://www.bio-oracle.org; (9) marine eco-regions: https://data.unep-wcmc.org/datasets/38; (10) fish trophic interactions: https://www.globalbioticinteractions.org; (11) reef fish abundance data: https://portal.aodn.org.au/search. (12) GASPAR dataset: https://rs.figshare.com/collections/Supplementary_material_from_Coral_reef_fishes_reveal_strong_divergence_in_the_prevalence_of_traits_along_with_the_global_diversity_gradient_/5647995/2. The list of coral-associated fish compiled from literature as well as the data used for the fish range validation and the dataset of functional traits are provided together with all scripts used in the analyses at https://doi.org/10.5281/zenodo.5702972[54].

## Code availability

All the scripts and data permitting to replicate the analyses and reproduce the figures are available from https://doi.org/10.5281/zenodo.5702972 [54].

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

## Acknowledgements
V.P. was supported by the Institut Universitaire de France (IUF), the BNP Paribas Foundation (Reef Services Project) and the French National Agency for Scientific Research (ANR; REEFLUX Project; ANR-17-CE32-0006). This research is also product of the SCORE-REEF group (G.S, V.P. and F.G.) funded by the Centre de Synthèse et d'Analyse sur la Biodiversité (CESAB) of the Foundation pour la Recherche sur la Biodiversité and the Agence Nationale de la Biodiversité. F.M. acknowledges the support of the Bertarelli Foundation. G.S. and P.S.A.B. performed part of the research in the context of the Exploratory Project EUReefs of the European Commission, Joint Research O.O. was funded by Academy of Finland (grant no. 309581), Jane and Aatos Erkko Foundation, Research Council of Norway through its Centres of Excellence Funding Scheme (223257), and the European Research Council (ERC) under the European Union's Horizon 2020 research and innovation programme (grant agreement No 856506; ERC-synergy project LIFEPLAN). The views expressed are purely those of the authors and may not in any circumstance be regarded as stating an official position of the European Commission. We thank Kevin Lafferty for providing useful comments and criticism on an early draft of the paper.

## Author contributions
Conceptualization: G.S. and V.P.; data curation, formal analysis, and visualization: G.S. and V.P.; G.S. wrote the original draft, with contributions from V.P., M.C., F.M., J.A.V., O.O., S.F., and P.S.A.B.; G.S. revised the manuscript, and P.S.A.B., M.C., S.F., F.G., F.M., S.M., O.O., S.P., J.A.V., and V.P. contributed to the final version.

## Competing interests
The authors declare no competing interests.

## Additional information

**Peer Review Information** *Nature Communications* thanks the anonymous reviewer(s) for their contribution to the peer review of this work. Peer reviewer reports are available.

