## [Peer Review File · Nature Communications]

Peer Review comments, first round review –

Reviewer #1 (Remarks to the Author):

This study evaluated the hypothesis that fish assemblages in remote locations may be more specialised and therefore vulnerable and this might translate into greater overall sustainability risk for these remote locations than would have previously been thought. The key element here is consideration of not just how the risk factors vary with distance from human impacts but critically, how aspects of the ecological system that underpin resilience may also vary along this gradient. A variety of global datasets on fish and coral presence/absence, environmental characteristics, hazards and species dependencies (interactions) were used. One key finding of the study was to highlight support for the notion that fish assemblages in remote areas may be more vulnerable to coral bleaching due to an increase in the overall dependency of fish assemblages on corals. They also used risk modelling to show that overall, these remote locations may be at much higher risk when these species dependences are considered than when they are not, potentially increasing global risk hotspots by 40%.

The conclusions of the study certainly provide fodder for deep consideration as they challenge many of the traditional notions about the resilience of remote, isolated and thus less impacted assemblages. Given this I think there would be a lot of scrutiny around the methodology that has led to these conclusions. I discuss several points along these lines below. But overall, I did find the study to be as clearly explained as is typically possible for such a massive, multi-step effort can be in the Nature Comms format. The objective of the study was clearly laid out and the methodology well linked to this. And of course, I'm quite sure the conclusions of this study would be of great interest to the global coral reef community, though elements of the general approach would be equally applicable to other similar situations where biodiversity is functionally dependent on habitat forming species which are vulnerable to global threats (kelp ecosystems for example).

Major comments

There are certainly a lot of assumptions made in this study but for me, the ones most critical to the overall findings are those to do with the coral dependencies and species interactions and how these are mapped to assemblages spatially. As a general statement, I worried about the degree to which the exclusion of various bits of data may have any spatial bias to it, especially with regards to the remoteness of reefs, given this remoteness factor was so integral to the hypotheses. I would imagine many remote reefs might be less well studied and this could translate into missing data about species occurrence (fish or coral) happening more often for these sites than less remote ones. The availability of data to support the interaction modelling was also of critical important. If remote sites tend to have a greater selection of unique species (endemic perhaps), are these equally well studied...enough to provide all the necessary data on trophic status and size that was required for the interaction modelling? Both of these factors could contribute to remote sites having 'simpler' trophic structures which may drive the elevated coral dependencies. So I think this needs to be addressed to assuage such concerns. One option would be to actually look at relationships between any data that had to be discarded and reef remoteness. You could also look at how TSS values vary depending on reef isolation, perusing the GASP dataset has similar levels of variation in this factor.

Related to the point above, I found the conclusion that coral-dependency increased with remoteness to be fascinating and wanted to hear a lot more about that. I don't think that finding gets the attention needed as there would be a lot to unpack there. A lot of trouble was gone to to include indirect effects. One might interpret this to mean the direct effects did not show a similar pattern on their own. Was this the case? What was actually driving this relationship? Theory and some empirical data suggest these remote sites may have greater functional redundancy? Was that the case here and if so, what then drives the increased coral dependency? Was it just a greater variety of specific coral-fish direct interactions? Or was it more about the indirect effects and the fish predator diversity? I feel this is a rather major result to leave with so little interpretation.

When using the random forest classifier to develop the species dependencies don't you need true absences for the training dataset? I fail to see how using the hypothesized absences can help provide you with a valid base case against which to formulate the predictive model.

How would consideration of actual abundance/density data change the conclusions here? Remote sites are typically observed to have much higher densities of many functional groups, especially higher trophic level ones. Would that increase or decrease the nature of the dependencies highlighted here and if the former, how do you then reconcile your overall conclusions against that.

The other major part of this study was the risk modelling. I found this was explained in only a very abstract manner which made it difficult to consider the ramifications of the assumptions. What are the values of these different inputs that are actually used and how do they lend themselves to being considered or not? It seems how you might weight the species dependency risks versus those of local and global hazards would make a big difference to the outcome when comparing the two scenarios. I thought these decisions should have been explained more clearly.

Minor Comments

Ln 88: There were a lot of caveats put on the language in discussing the results and this sentence sums them all up quite well..." preliminary support". Whilst I did appreciate and agree with this choice of language, I did find myself wondering if this level of confidence in the result was appropriate for a study in this high-profile journal.

Ln 422: It's not quite true that the pattern for the scaled-up relationship was the same as the average for individual species. The scaled-up graph shows the best fit to be pretty much flat after the drop while the one without it sloped upwards. And it was this upward slope that was featured in the results. So, what does it mean that it is not present in this scaled-up analysis?

Reviewer #2 (Remarks to the Author):

This is an exciting and well written article about the "validity" of the relationship between remoteness (the distance of coral reef from human settlement) and ecological specialization in a context of accelerated environmental changes. The global data set, the scale of analysis and the combination of theoretical and empirical approaches makes the study very interesting.

Briefly, authors point that theoretical evidence showed that ecological networks evolved towards increasing ecological specialization as they remained undisturbed. That is, ecological networks appear robust to extinctions, possibly due to consumers' tendency to specialize on temporally stable

resources. However, modifications to the resource conditions under which the network has evolved, determine that species have no chance/time to adapt, and so they could be not able to avoid local extinctions. In a context of accelerated environmental changes mainly due to anthropic effects (Anthropocene), these changes may collapse otherwise robust natural ecosystems.

In this context, authors suggest that more remote or isolated communities, in areas far from the effects of human wherein a high level of specialization was achieved, ecological networks would be more vulnerable to new environmental challenges, as it is expected under global change, promoting the ecosystem collapse compared to ecosystems in areas closer to humans. Assuming that remoteness of an ecosystem can limit the probability of external perturbations, they propose to evaluate the validity of the remoteness-specialization hypothesis, and whether a positive relationship between ecological specialization and remoteness may lead to a high risk of species extinction in remote ecosystems. I think the authors give support to their results. However, I have some concerns that I hope could improve the results and main discussion.

I think the introduction should be supported by more bibliography. The main proposed problem is supported by only one theoretical study (Strona and Lafferty 2016), other studies should be cited, for example Poniso et al 2019. In this line, authors should try to better support the connection between ecological specialization, ecosystem isolation and environmental stability. Maybe they could present a conceptual diagram, highlighting the main variables involved, and their potential connection.

Also, my main concern is that authors associate the remoteness of an ecosystem with environmental stability. However, the remoteness of an ecosystem determines the flow of individuals that a community received, and this could affect the degree of specialization. The flow of individuals has been recognized as a main determinant of species assembly and so of ecological networks. In this line, a high flow of individuals in more connected communities could reduce the effect of environmental change on the specialization in ecological networks.

In addition, more remoteness communities, because of a reduced flow of individuals (not associated with the stability of the environment) could have low species richness, and this could increase the extinction risk.

I would like the authors incorporate in the discussion and maybe in the introduction the size (species richness) of the ecological networks, and the potential flow that a community could receive according to its degree of geographic isolation. This could be great considering the data set used has a notably a wide isolation-centrality gradient.

The results represent an empirical evidence of the remoteness-specialization hypothesis. I think it is an important contribution of this article and could be more emphasized

Reviewer #1 (Remarks to the Author):

This study evaluated the hypothesis that fish assemblages in remote locations may be more specialised and therefore vulnerable and this might translate into greater overall sustainability risk for these remote locations than would have previously been thought. The key element here is consideration of not just how the risk factors vary with distance from human impacts but critically, how aspects of the ecological system that underpin resilience may also vary along this gradient. A variety of global datasets on fish and coral presence/absence, environmental characteristics, hazards and species dependencies (interactions) were used. One key finding of the study was to highlight support for the notion that fish assemblages in remote areas may be more vulnerable to coral bleaching due to an increase in the overall dependency of fish assemblages on corals. They also used risk modelling to show that overall, these remote locations may be at much higher risk when these species dependences are considered than when they are not, potentially increasing global risk hotspots by 40%.

The conclusions of the study certainly provide fodder for deep consideration as they challenge many of the traditional notions about the resilience of remote, isolated and thus less impacted assemblages. Given this I think there would be a lot of scrutiny around the methodology that has led to these conclusions. I discuss several points along these lines below. But overall, I did find the study to be as clearly explained as is typically possible for such a massive, multi-step effort can be in the Nature Comms format. The objective of the study was clearly laid out and the methodology well linked to this. And of course, I'm quite sure the conclusions of this study would be of great interest to the global coral reef community, though elements of the general approach would be equally applicable to other similar situations where biodiversity is functionally dependent on habitat forming species which are vulnerable to global threats (kelp ecosystems for example).

RESPONSE: Thank you very much for the positive feedback. We also believe that our hypotheses and findings might extend far beyond coral-reefs and have general ecological validity and relevance for conservation.

Major comments

There are certainly a lot of assumptions made in this study but for me, the ones most critical to the overall findings are those to do with the coral dependencies and species interactions and how these are mapped to assemblages spatially. As a general statement, I worried about the degree to which the exclusion of various bits of data may have any spatial bias to it, especially with regards to the remoteness of reefs, given this remoteness factor was so integral to the hypotheses. I would imagine many remote reefs might be less well studied and this could translate into missing data about species occurrence (fish or coral) happening more often for these sites than less remote ones. The availability of data to support the interaction modelling was also of critical important. If remote sites tend to have a greater selection of unique species (endemic perhaps), are these equally well studied...enough to provide all the necessary data on trophic status and size that was required for the interaction modelling? Both of these factors could contribute to remote sites having 'simpler' trophic structures which may drive the elevated coral dependencies. So I think this needs to be addressed to assuage such concerns. One option would be to actually look at relationships between any data that had to be discarded and reef remoteness. You could also look at how TSS values vary depending on reef isolation, perusing the GASP dataset has similar levels of variation in this factor.

RESPONSE: We have now explored this aspect from different perspectives to ensure the validity of our conclusions. We have now compared the ratio between number of "raw" occurrence records (from GBIF and OBIS) vs. number of species per locality vs. remoteness finding no relationship (see Extended Data

Fig. 2a). We have also explored thoroughly if and how remoteness affects the availability of ecological information. In particular, we computed, for each locality, the fraction of species for which we had enough ecological information to conduct all of our analyses. Then we explored if such value is negatively affected by remoteness. Again, we found that remoteness does not affect the availability of ecological data (Extended Data Fig. 2d). As suggested, we also compared remoteness with TSSs obtained by comparing the predicted set of species vs. the set of species reported by GASPAR. Again, we found no relationship (Extended Data Fig. 2b). We also used the individual species' TSS (obtained by comparing, for each species, the observed occurrences from GASPAR dataset, and the ranges we obtained from the occurrence data), to check whether we had "better" geographical information on coral associated species compared to the non-associated ones. We found that this is not the case: the average TSSs for the associated and non-associated ones are almost identical (0.40 ± 0.29 and 0.41 ± 0.33 respectively). We also found that the average species-specific TSSs of fish species in each reef locality do not vary with remoteness (Extended Data Fig. 2c).

Related to the point above, I found the conclusion that coral-dependency increased with remoteness to be fascinating and wanted to hear a lot more about that. I don't think that finding gets the attention needed as there would be a lot to unpack there. A lot of trouble was gone to to include indirect effects. One might interpret this to mean the direct effects did not show a similar pattern on their own. Was this the case? What was actually driving this relationship?

RESPONSE: This is a good point. Building the networks was indeed a complex exercise but, in the end, we found that the overall dependency (i.e. combining direct fish-coral links and indirect links to corals across fish trophic interactions) is strongly driven by the prevalence of direct links. Therefore, at first we had not emphasized this result as not particularly interesting, while we had left the network part in the paper as it was – at least – showing that the pattern observed for direct dependencies remains consistent when also indirect dependencies are considered. However, stimulated by the Reviewer's remark, we have now explored more in depth the issue. In particular, we have looked at how the relative importance of direct links compared to indirect links to coral varied with remoteness. This analysis revealed that not only with remoteness we have more coral dependent fish, but also that the relative importance of strong, direct coral dependencies increases with distance from human settlements. This further strengthens our results and conclusions, as, with remoteness, we see an increase not only in the fraction of fish diversity potentially affected by coral loss increases with remoteness, but also in the intensity of the potential effect of coral loss on associated fish.

Theory and some empirical data suggest these remote sites may have greater functional redundancy? Was that the case here and if so, what then drives the increased coral dependency?

RESPONSE: we have now checked the potential relationship between remoteness and functional redundancy, by using the method (and functional diversity datasets) as in Mouillot et al. (2014; Proc. Natl. Acad. Sci. USA 111, 13757), identifying, for each locality, the number of unique functional entities, and comparing such number with remoteness. We found no relationship (Extended Data Fig. 2f), which suggests that functional redundancy does not significantly vary with distance from humans and cannot be therefore considered a confounding factor in our analyses.

Was it just a greater variety of specific coral-fish direct interactions? Or was it more about the indirect effects and the fish predator diversity? I feel this is a rather major result to leave with so little interpretation.

RESPONSE: As explained above, our new analysis reveal that the relative contributions of direct and indirect interactions vary with remoteness, with direct interactions becoming more and more important with distance from human settlements compared to indirect interactions (see the new Fig. 3).

When using the random forest classifier to develop the species dependencies don't you need true absences for the training dataset? I fail to see how using the hypothesized absences can help provide you with a valid base case against which to formulate the predictive model.

RESPONSE: we have now better clarified this aspect in the Method section. One problem with available data on fish trophic interactions is that there is virtually no existing dataset providing true absences. The trophic interactions not documented in our large datasets for the included species (i.e. the “missing” pairwise combinations of the species for which some interaction is present in the dataset) provide a good starting point. However, the large set of missing pairwise combinations is quite likely to include at least some possible/existing interactions. This might add much noise to the data, and lead to a poorly performing model. For this, we narrowed down the set by identifying among the candidate absences, a subset having high probability of being true absences (i.e. identifying combinations where the potential predator was of low trophic level and/or much smaller than the prey). This step dramatically improved our model, leading to an exceptional precision and accuracy (TSS = 0.93; type I error rate = 0.05; type II error rate = 0.02 from the comparison of a training and testing sets including each 50% of the interactions from GLOBI), which makes a very solid case supporting our approach.

How would consideration of actual abundance/density data change the conclusions here? Remote sites are typically observed to have much higher densities of many functional groups, especially higher trophic level ones. Would that increase or decrease the nature of the dependencies highlighted here and if the former, how do you then reconcile your overall conclusions against that.

RESPONSE: This is an interesting point. Obtaining data on fish abundances comparable (in term of spatial and taxonomic coverage) with the occurrence data we have used in our analyses is not possible. However, we have now explored the issue using all the data available from the Reef Life Survey (RLS) dataset. RLS data are not directly comparable with ours for multiple reasons. The most important are that RLS data geographical coverage consists of a set of point locations; and that the occurrences it reports are only snapshots of the local fish assemblages, while our dataset aims at being as much complete as possible in describing local fish diversity. Nevertheless, RLS data offer a valuable benchmark to explore whether our conclusions can change when considering local species abundances. We therefore compared the abundances of coral associated vs. non associated fish across all the surveys in the RLS dataset, and then we explored if the ratio between the local abundance of associated fish vs. non associated fish varied with remoteness. We found that, on average, coral associated fish are more abundant than the other species (with an average number of individuals per survey of 782 for coral associated species vs. 658 for non associated species). Furthermore, the local proportion of associated individuals is unaffected by remoteness (Extended Data Fig. 2e). Thus, our conclusions hold both when considering species occurrences and species abundances.

The other major part of this study was the risk modelling. I found this was explained in only a very abstract manner which made it difficult to consider the ramifications of the assumptions. What are the values of these

different inputs that are actually used and how do they lend themselves to being considered or not? It seems how you might weight the species dependency risks versus those of local and global hazards would make a big difference to the outcome when comparing the two scenarios. I thought these decisions should have been explained more clearly.

RESPONSE: the risk assessment framework we describe in the paper is indeed abstract (we actually referred to it as “illustrative”). In our purpose, the framework was meant to illustrate in a formal way how to account for ecological dependencies when trying to quantify species risk of going locally extinct. The Reviewer’s remark is correct: as we have explained in the section “Assumptions of the risk assessment equations” the weights given to the different parameters can drive the overall risk patterns. We have already explored some of these aspects in the previous version of the MS (particularly, exploring thoroughly the assumption of constant vulnerability of fish assemblages to local and global hazards). In addition to that, we have now modified the risk equation with the addition of two parameters weighting the relative importance of human impacts vs. ecological dependencies, and explored how the patterns (which we now measure, for better clarity, as the trend lines’ slopes of the risk-remoteness relationship) change when playing with weight parameters. In this way we are ideally showing the full spectrum of possible trends that the risk-remoteness relationship can assume. Identifying the “true” weights is clearly a challenging question that we cannot answer in this paper. The decision we took in the previous version, i.e. assigning equal weights to the global+local hazards and ecological vulnerability seems a reasonable compromise. In practice, this means that we consider that a fish is equally endangered by being subjected to high fish pressure or by being a coral-dependent species in a locality highly susceptible to coral bleaching. Also, we understand the Reviewer’s statement that the risk assessment framework appears as a major part of our study but, in reality, we devised it quite late in the process, as a useful conceptual tool to better formalize our theory. The map showing the remarkable segregation in hotspots of risk stemming from local + global impacts, and hotspots stemming from ecological dependency is actually our major finding, as it clearly support our claim that there is no safe place for biodiversity in the Anthropocene. Therefore, we have now anticipated that part, and reworded the risk assessment part, doing our best to clarify all of these aspects.

Minor Comments

Ln 88: There were a lot of caveats put on the language in discussing the results and this sentence sums them all up quite well...” preliminary support”. Whilst I did appreciate and agree with this choice of language, I did find myself wondering if this level of confidence in the result was appropriate for a study in this high-profile journal.

RESPONSE: Thanks for the comment. Also thanks to all the additional analyses providing strong support to our findings, we have reworked the text and removed/replaced expressions potentially weakening our message with more confident statements.

Ln 422: It’s not quite true that the pattern for the scaled-up relationship was the same as the average for individual species. The scaled-up graph shows the best fit to be pretty much flat after the drop while the one without it sloped upwards. And it was this upward slope that was featured in the results. So, what does it mean that it is not present in this scaled-up analysis?

RESPONSE: Here we referred to the idea that remoteness flattens the risk-remoteness relationship, which

can be appreciated both when focusing on individual species risk and on the total risk for fish assemblages. The small difference in slopes in the piecewise regression was due by the negative relationship between diversity and remoteness, which attenuates the “raw” risk (total number of species potentially at risk) as we move away from human influence. With the new data, now the differences are even more attenuated. We have reported in the main text the results referring to the average risk as our aim is that of showing the general ecological mechanism which increases the dependencies with remoteness; then, of course, such effect can combine with local species richness generating different scenarios of risk, but the core of our message is that the dependency will increase, which refers to the degree to which each species depends on corals. We have now clarified this aspect, also adding a plot of fish diversity vs. remoteness (Extended Data Fig. 4a).

Reviewer #2 (Remarks to the Author):

This is an exciting and well written article about the “validity” of the relationship between remoteness (the distance of coral reef from human settlement) and ecological specialization in a context of accelerated environmental changes. The global data set, the scale of analysis and the combination of theoretical and empirical approaches makes the study very interesting. Briefly, authors point that theoretical evidence showed that ecological networks evolved towards increasing ecological specialization as they remained undisturbed. That is, ecological networks appear robust to extinctions, possibly due to consumers’ tendency to specialize on temporally stable resources. However, modifications to the resource conditions under which the network has evolved, determine that species have no chance/time to adapt, and so they could be not able to avoid local extinctions. In a context of accelerated environmental changes mainly due to anthropic effects (Anthropocene), these changes may collapse otherwise robust natural ecosystems. In this context, authors suggest that more remote or isolated communities, in areas far from the effects of human wherein a high level of specialization was achieved, ecological networks would be more vulnerable to new environmental challenges, as it is expected under global change, promoting the ecosystem collapse compared to ecosystems in areas closer to humans. Assuming that remoteness of an ecosystem can limit the probability of external perturbations, they propose to evaluate the validity of the remoteness-specialization hypothesis, and whether a positive relationship between ecological specialization and remoteness may lead to a high risk of species extinction in remote ecosystems. I think the authors give support to their results. However, I have some concerns that I hope could improve the results and main discussion.

RESPONSE: Thank you very much for the positive feedback and the useful comments.

I think the introduction should be supported by more bibliography. The main proposed problem is supported by only one theoretical study (Strona and Lafferty 2016), other studies should be cited, for example Poniso et al 2019. In this line, authors should try to better support the connection between ecological specialization, ecosystem isolation and environmental stability. Maybe they could present a conceptual diagram, highlighting the main variables involved, and their potential connection.

RESPONSE: Thanks for this remark. We have read carefully the suggested paper, which is indeed much relevant to our hypothesis, and cited it in the Introduction alongside with additional references supporting the idea that isolation and disturbances can affect the degree of specialization in natural communities.

Also, my main concern is that authors associate the remoteness of an ecosystem with environmental stability. However, the remoteness of an ecosystem determines the flow of individuals that a community received, and this

could affect the degree of specialization. The flow of individuals has been recognized as a main determinant of species assembly and so of ecological networks. In this line, a high flow of individuals in more connected communities could reduce the effect of environmental change on the specialization in ecological networks. In addition, more remoteness communities, because of a reduced flow of individuals (not associated with the stability of the environment) could have low species richness, and this could increase the extinction risk. I would like the authors incorporate in the discussion and maybe in the introduction the size (species richness) of the ecological networks, and the potential flow that a community could receive according to its degree of geographic isolation. This could be great considering the data set used has a notably a wide isolation-centrality gradient.

RESPONSE: Thanks for this insightful remark. The Reviewers' remarks are correct, in that from an evolutionary perspective, individual flow modulated by isolation (from a purely geographical meaning, as in theory of Island Biogeography) can both affect the evolution of specialization (that would be more likely with a low flow – i.e. in more isolated localities) and species extinction risk. We have now mentioned this aspect in the Introduction. However, we have also reworked the text to better clarify that our measure of remoteness – travel distance from major human settlements – ideally serves as a proxy for both direct human impacts on local communities and isolation. That is, although remoteness correlates with isolation (as we now show in Extended Data Fig.1) it is in fact a more complex and comprehensive measure and a natural choice for testing our hypotheses.

The results represent an empirical evidence of the remoteness-specialization hypothesis. I think it is an important contribution of this article and could be more emphasized

Authors: thanks, we have now better emphasized this aspect when first introducing the relationship between fish-coral associations and remoteness.

Peer Review comments, second round review –

Reviewer #1 (Remarks to the Author):

I thank the authors for their very thorough work to address the concerns raised by myself and the other reviewer. Having reviewed the revised manuscript and associated supplemental analyses, I feel much more confident that conclusions offered are well defended in a rigorous and quantitative manner. As I said before, my main concern was any potential bias in the data availability due to the remote and unique nature of the systems that are the focus here. But based on the new analysis provided, I don't think this is a concern. Given the conclusions here are likely to raise discussion, I think these added analysis will build confidence in them. I have no further comments to add.

Reviewer #2 (Remarks to the Author):

I have read the authors responses and the revision of this manuscript, and I think that, the authors have adequately addressed all my concerns. Overall, it is much improved. The introduction and conceptual framing are much better developed and supported by more bibliography, as it is the discussion.

Reviewer #1 (Remarks to the Author):

I thank the authors for their very thorough work to address the concerns raised by myself and the other reviewer. Having reviewed the revised manuscript and associated supplemental analyses, I feel much more confident that conclusions offered are well defended in a rigorous and quantitative manner. As I said before, my main concern was any potential bias in the data availability due to the remote and unique nature of the systems that are the focus here. But based on the new analysis provided, I don't think this is a concern. Given the conclusions here are likely to raise discussion, I think these added analysis will build confidence in them. I have no further comments to add.

Reviewer #2 (Remarks to the Author):

I have read the authors responses and the revision of this manuscript, and I think that, the authors have adequately addressed all my concerns. Overall, it is much improved. The introduction and conceptual framing are much better developed and supported by more bibliography, as it is the discussion.

Authors: We thank both Reviewers for their constructive criticism which has greatly improved the quality of our paper.